# Near-Infrared (NIR) Silver Sulfide (Ag_2_S) Semiconductor Photocatalyst Film for Degradation of Methylene Blue Solution

**DOI:** 10.3390/ma16010437

**Published:** 2023-01-03

**Authors:** Zahrah Ramadlan Mubarokah, Norsuria Mahmed, Mohd Natashah Norizan, Ili Salwani Mohamad, Mohd Mustafa Al Bakri Abdullah, Katarzyna Błoch, Marcin Nabiałek, Madalina Simona Baltatu, Andrei Victor Sandu, Petrica Vizureanu

**Affiliations:** 1Faculty of Chemical Engineering & Technology, Universiti Malaysia Perlis (UniMAP), Arau 01000, Malaysia; 2Centre of Excellence Geopolymer and Green Technology (CEGeoGTech), Universiti Malaysia Perlis (UniMAP), Arau 01000, Malaysia; 3Faculty of Electronic Engineering & Technology, Universiti Malaysia Perlis (UniMAP), Arau 02600, Malaysia; 4Faculty of Mechanical Engineering and Computer Science, Częstochowa University of Technology, 42-201 Częstochowa, Poland; 5Department of Technologies and Equipments for Materials Processing, Faculty of Materials Science and Engineering, Gheorghe Asachi Technical University of Iaşi, Blvd. Mangeron, No. 51, 700050 Iasi, Romania; 6National Institute for Research and Development in Environmental Protection INCDPM, Splaiul Independentei 294, 060031 Bucharest, Romania; 7Romanian Inventors Forum, Str. Sf. P. Movila 3, 700089 Iasi, Romania; 8Technical Sciences Academy of Romania, Dacia Blvd 26, 030167 Bucharest, Romania

**Keywords:** near-infrared irradiation, silver sulfide, cellulose film, photocatalysis, methylene blue

## Abstract

A silver sulfide (Ag_2_S) semiconductor photocatalyst film has been successfully synthesized using a solution casting method. To produce the photocatalyst films, two types of Ag_2_S powder were used: a commercialized and synthesized powder. For the commercialized powder (CF/comAg_2_S), the Ag_2_S underwent a rarefaction process to reduce its crystallite size from 52 nm to 10 nm, followed by incorporation into microcrystalline cellulose using a solution casting method under the presence of an alkaline/urea solution. A similar process was applied to the synthesized Ag_2_S powder (CF/syntAg_2_S), resulting from the co-precipitation process of silver nitrate (AgNO_3_) and thiourea. The prepared photocatalyst films and their photocatalytic efficiency were characterized by Fourier transform infrared spectroscopy (FTIR), X-ray diffraction (XRD), and UV-visible spectroscopy (UV-Vis). The results showed that the incorporation of the Ag_2_S powder into the cellulose films could reduce the peak intensity of the oxygen-containing functional group, which indicated the formation of a composite film. The study of the crystal structure confirmed that all of the as-prepared samples featured a monoclinic acanthite Ag_2_S structure with space group P_21_/C. It was found that the degradation rate of the methylene blue dye reached 100% within 2 h under sunlight exposure when using CF/comAg_2_S and 98.6% for the CF/syntAg_2_S photocatalyst film, and only 48.1% for the bare Ag_2_S powder. For the non-exposure sunlight samples, the degradation rate of only 33–35% indicated the importance of the semiconductor near-infrared (NIR) Ag_2_S photocatalyst used.

## 1. Introduction

Since the prehistoric age, humans have used dyes from natural resources for coloring purposes. The lengthy process, poor colorfastness, excessive cost of producing natural dyes, and the increasing demand for textiles led to the discovery of synthetic dyes from petroleum compounds that outperformed the properties of natural dyes [1,2,3]. Statistically, more than 7 × 10^5^ tons of synthetic dyes are produced worldwide, and about 1 × 10^4^ of them are used in industry [4]. However, these synthetic dyes have a negative impact on the environment, such as water pollution. Textile industries discharge around 35% of dye wastewater into water bodies as effluent annually [5]. These dyes can contaminate the aquatic habitat, which may enter the food chain [6,7]. Methylene blue (MB), or C_16_H_18_N_3_SCl, is one of the synthetic dye materials used in textile industries. When dissolved in water, it results in the formation of a blue-colored solution. The discharging of MB dye into the environment is considered as a significant threat for aesthetical and toxicological reasons. In terms of aquatic life, its existence in the water could reduce sunlight transmittance and decrease oxygen solubility due to the high molar absorption coefficient (~8.4 × 104 L·mol^−1^·cm^−1^ at 664 nm) of MB dyes [8,9,10]. At a certain concentration, it can cause serious threats to human health, including respiratory distress, abdominal disorders, blindness, and digestive and mental disorders [11].

In recent years, tremendous attempts have been made to improve industrial wastewater treatment methods. These methods can be divided into physical [12,13,14,15,16], biological [17,18,19,20,21], and chemical methods. For the chemical methods, there are several conventional chemical dye-removal processes that have been used, e.g., the coagulant flocculation, electrochemical destruction, advanced oxidation process (AOP), etc. [22,23,24,25]. Amongst these, the advanced oxidation processes (AOPs) have received the most attention due to their significant competence in acting toward a wide range of organic or inorganic dye pollutants in the aqueous phase, by converting those pollutants into stable inorganic compounds such as water, carbon dioxide, and salt without any footprint and sludge production [26,27]. The AOPs also rely on the in situ production of greatly reactive hydroxyl radicals (OH•) [28].

In the midst of the AOPs, heterogeneous photocatalysis has been considered a leading method and desired breakthrough to cope with organic contaminants. Photocatalysis is a reaction that involves light (photoreaction) and accelerates the reaction due to the presence of a catalyst that absorbs light energy to form the reducing and oxidizing (electron–hole pairs) ions on the surface of the catalyst. The electron’s handover occurs during the oxidation–reduction process [29]. The reduction of an acceptor occurs when the electrons (*e*^−^) combine with oxygen in the water to generate an anion (O_2_^−^), which oxidizes the hydroxyl radical (OH•), while the hole (*h^+^*) will oxidize the dissolved hydroxyl and convert it into a radical with great energy [30]. These processes work simultaneously. The photocatalytic properties of various metal oxides (e.g., titania, TiO_2_, and zinc oxide, ZnO) have been extensively studied. The results of these studies show that the metal oxides become active potential photocatalysts in wastewater treatment [31,32,33,34]. However, due to a wide bandgap, they can only absorb the ultraviolet (UV) light and show poor performance in the visible–near-infrared (NIR) range of the solar spectrum [29,35,36,37]. In addition, the solar spectrum itself is dominated by the visible (46%) and NIR spectral (49%) rather than the UV spectral (5%) [38]. In addition, the manufacturing cost is relatively high for many of the metal oxide materials [39,40]. Therefore, metal sulfides have become an option.

Among all of the metal sulfide photocatalyst materials, silver sulfide (Ag_2_S) shows more efficient charge separation, attributed to the remarkable synergistic effects of strong NIR light absorption and excellent surface properties [41,42]. Furthermore, compared to the other metal sulfide compounds such as cadmium selenide, CdS, and lead selenide, PbS, Ag_2_S is found to be acceptable for wastewater treatment due to its non-toxic properties [43,44,45]. Silver sulfide has an eminent performance in the degradation of pollutants, solar energy conversion, and production of hydrogen with various approaches that have been well established for synthesizing a variety of visible–NIR-light driven Ag_2_S photocatalysts. For example, Ag_2_S synthesized using the facile ion-exchange method at room temperature can be used as an effective photocatalyst for the decomposition of methylene orange (MO). The results of a study confirmed the excellent photo-oxidation performance of Ag_2_S since MO can be completely photodegraded with Ag_2_S in just 30 min, and 70 min under visible light and NIR light irradiation. This performance is due to the narrow band gap of Ag_2_S, 1.078 eV, and the lower recombination efficiency and photogenerated electron–hole pairs of Ag_2_S during the photocatalytic process [46]. The Ag_2_S photocatalyst has usually been used in the powder form [43,47,48,49]. This is because the photocatalyst powder can be well dispersed in suspensions [50].

However, this photocatalyst powder exhibits certain drawbacks especially for nano-sized powders/particles. During the degradation process, the photocatalyst may undergo coagulation due to the instability of the nano-sized particles, which will hamper the light incidence on the active centers, consequently reducing its catalytic activity [51]. Furthermore, for the slurry system, the main challenge is to recover the nano-sized photocatalyst particles from the treated water [52,53,54]. This condition leads to the requirements of high filtration costs of catalyst removal [55], hindering its industrial application and impractical. In addition, the Ag_2_S nanoparticles are susceptible to photo-corrosion, which means that the sulfide ions have the potential to be oxidized into sulfur by photogenerated holes when they are exposed to irradiation [56]. There have been a lot of efforts towards increasing the photocatalytic efficiency and preventing the photo-corrosion of Ag_2_S [57,58,59,60]. A photocatalyst film is one of the plausible material technologies to diminish the impacts of photocatalyst powder on the environment [61]. A photocatalyst film has promising performance in terms of adsorption and efficient mass transport. The adsorption ability is maxed out by the larger surface area that allows more catalyst deposit in the film, resulting in higher photocatalytic activity. Moreover, the mass transport of reactants, intermediates, and products is enhanced by the compact substrate that allows better contact with the catalyst, and prevents the leakage of the powder of the photocatalyst material into clean water [62,63]. Table 1 shows the cost comparison for some types of photocatalyst, especially titania (TiO_2_), which is a common photocatalyst used for industrial purposes. Those types of photocatalyst are used under a UV lamp-assisted reactor for the photocatalytic processes and consume a certain amount of energy (power). For an industrial batch process, it will consume a lot of energy with a higher cost of operation. Thus, our study introduces a composite film (cellulose/Ag_2_S) that can effectively degrade methylene blue concentrations using only sunlight, which is cost-effective. Furthermore, the composite film can also be easily removed from the water after the treatment process without a costly separation method.

To date, bio-based products such as cellulose film have become attractive compounds due to their excellent properties that are harmless to the environment. A cellulose film has distinctive characteristics such as transparency, robustness, low water content, etc. These properties lead to the wide usage of cellulose film as a renewable alternative to petroleum-based materials [68]. Furthermore, it is hypothesized that cellulose may become one of the solutions to improve the efficiency of electron donors and to establish the hole (*h*^+^) transporter during the light irradiation process. To the best of our knowledge, the incorporation of Ag_2_S powder into cellulose film has not been studied, although the integration between semiconductor catalyst and cellulose film is a promising innovation for photocatalytic applications. Therefore, the integration of Ag_2_S in cellulose-derived film is hoped to enhance the photocatalytic efficiency. Thus, this study aimed to synthesize cellulose/Ag_2_S film in the presence of an alkali/urea solution using a simple solution casting method to describe the transformation of the functional group and the structure of the cellulose thin film due to the deposition of Ag_2_S. We also compared the photocatalytic efficiency of the commercial Ag_2_S and synthesized Ag_2_S doped into the cellulose film via the degradation of the MB solution.

## 2. Materials and Methods

The current study successfully synthesized films that were stated as CFs for cellulose film, cellulose films with commercial Ag_2_S (CF/comAg_2_S), and cellulose films with synthesized Ag_2_S (CF/syntAg_2_S). The materials and methods used for synthesizing the Ag_2_S photocatalyst film are described in the section below.

### 2.1. Materials and Reagents

The materials and reagents used consist of microcrystalline cellulose, MCC (≤100%), commercial silver sulfide, Ag_2_S, silver nitrate, AgNO_3_, thiourea, (NH_2_)_2_CS, and polyvinyl alcohol, PVA. All of these materials were bought from Sigma-Aldrich. The PVA was dissolved with distilled water at the temperature of 90 °C until the transparent PVA solution with a concentration of 5.0% (*w*/*w*) occurred. Chemicals including sodium hydroxide (NaOH), glycerol (C_3_H_8_O_3_), and acetone (C_3_H_6_O) were supplied by HmbG chemicals. The acetone was diluted in distilled water with a ratio of 2:1 as the agent of regeneration for cellulose films, called an acetone bath. Urea (CO(NH_2_)_2_) was obtained from Bendosen Laboratory Chemicals.

### 2.2. Preparation of Sodium Hydroxide/Urea (NaOH/urea) Aqueous Solution

To obtain the NaOH/urea solution, the NaOH pellets and urea powder were weighed to obtain 7 g and 12 g of mass, respectively [47,69]. Subsequently, each sample of NaOH and urea was diluted in a separate beaker glass before being mixed in a 100 mL volumetric flask. The balanced chemical reaction worked in the mixture can be written as:2NaOH(aq) + CO(NH_2_)_2_(aq) → 2NH_3_(g) + Na_2_CO_3_(aq)(1)

The solution was agitated to obtain a homogenous solution. Once the homogeneity was obtained, the solution was stored for the next process.

### 2.3. Liquid Phase Rarefaction of Commercial Ag_2_S by Magnetic Stirring Technique

Prior to the incorporation of the commercial Ag_2_S into the cellulose film, it was essential to break down the bulk Ag_2_S into smaller particle sizes. In this step, the commercial Ag_2_S powder was weighed according to the mass variation that is displayed in Table 2.

After the desired amount of Ag_2_S was obtained, the powder was placed into a beaker containing 6 mL of distilled water and vigorously stirred using a magnetic stirrer under ambient temperature for 24 h. The crystallite size of rarefaction Ag_2_S was then investigated using X-ray diffraction (XRD) to confirm its size reduction.

### 2.4. Synthesis of the Cellulose Film

The cellulose films were prepared using a solution casting method. The cellulose with NaOH/urea aqueous solution was used as the main reagent to synthesize the cellulose films. Both commercial Ag_2_S and synthesized Ag_2_S were incorporated into the cellulose system to study the different behavior of the cellulose film with commercial and synthesized Ag_2_S. The composition of NaOH, urea, and water was different for each process as shown in Table 3. Figure 1 shows the overall schematic procedure to synthesize the CFs, CF/comAg_2_S, and CF/syntAg_2_S samples.

#### 2.4.1. Synthesis Cellulose Film (CF) and Cellulose Film/Commercial Ag_2_S (CF/comAg_2_S)

At the beginning of the process, 15 mL of NaOH/urea aqueous solution was precooled to a temperature of ~1 °C in the ice bath. After that, 3 wt% of microcrystalline cellulose (MCC) was added into the precooled solvent, and the mixture was rapidly agitated for 20 min. Then, the cellulose solution was taken out from the ice bath and 2.5 wt% of glycerol [70] was wisely dropped into the solution. Afterwards, the solution was spun for another hour to achieve the homogeneity. Once the dope and viscous solution was obtained, it was cast onto an 8.5 cm × 6 cm × 0.2 cm glass mold. Subsequently, the casted sample was regenerated in an acetone coagulant bath [71] until opaque-sheet-like cellulose hydrogel was formed. The hydrogel was soaked in distilled water for neutralization. Before drying, the hydrogel was submerged in a 5 wt% polyvinyl alcohol (PVA) bath for 2 h [72]. Finally, a transparent cellulose film was obtained. The same steps were also conducted in order to synthesize the CF/comAg_2_S specimens, except the 6mL of refracted Ag_2_S produced from Section 2.3 was first poured into the NaOH/urea solution prior to the addition of the desired amount of MCC into the precooled solution.

#### 2.4.2. Synthesis of Cellulose Film/Synthesized Ag_2_S (CF/syntAg_2_S)

The co-precipitation method was used to synthesize the Ag_2_S particles. Silver nitrate and thiourea were the main precursors to produce the Ag_2_S slurry. The presence of silver and sulfide ions in the solution was required for the chemical deposition process to occur in order to complete the formation of Ag2S. The chemical reaction involved during the deposition process is shown in the following formulas [73,74]:(NH_2_)_2_CS + H_2_O ↔ (NH_2_)_2_CO + H_2_S(2)
2AgNO_3_ + H_2_S → Ag_2_S + 2HNO_3_(3)

Table 3 shows the final composition that was utilized in this section for the whole synthesis process:

**Table 3 materials-16-00437-t003:** Sample code and sample composition to synthesize samples CF/syntAg_2_S.

Sample Code	Molarity of AgNO_3_	AgNO_3_ Solution		Thiourea Solution			
		AgNO_3_	Water	NaOH	Thiourea	Urea	Water
CF/syntAg_2_S1	0.1 M	0.084 g	5.0 mL	1.6 g	1.3 g	1.6 g	10.5 mL
CF/syntAg_2_S2	0.3 M	0.254 g	5.0 mL	1.6 g	1.3 g	1.6 g	10.5 mL
CF/syntAg_2_S3	0.5 M	0.422 g	5.0 mL	1.6 g	1.3 g	1.6 g	10.5 mL

According to Table 3, the total volume of solution that was used in the synthesis of CF/syntAg_2_S samples was 20 mL (5 mL AgNO_3_ solution and 15 mL thiourea solution). The solutions were prepared in two different beakers. At the beginning of the process, 15 mL of thiourea solution was poured into a 50 mL glass beaker while stirring on a magnetic stirrer. After that, the 5 mL of AgNO_3_ solution was added dropwise into the stirred thiourea solution. At this stage, a black precipitate was obtained, which indicated the formation of Ag_2_S particles. The solution was left to stir for another 15 min to ensure all of the substances completely reacted. Then, the precipitated solution was precooled in the ice bath for 20 min. Subsequently, 3% *w*/*v* of MCC powder was added into the black cold-precipitated solution and continuously stirred until the MCC was well dispersed in the solution. The mixed solution was then removed from the ice bath, dripped with glycerol, and spun for an entire hour. The cellulose/synthesized AgNO_3_ solution was transferred into the glass mold and then regenerated by using an acetone bath and PVA bath, in sequence. Lastly, the cellulose film obtained after the regenerated film was open-air dried for 2 days. All of the procedures above were repeated for different molarities of synthesized AgNO_3_.

### 2.5. Photocatalytic Activity Test

For photocatalytic testing, a concentration of 10 ppm of methylene blue (MB) solution was prepared. An amount of 0.1 g of MB powder was diluted with distilled water in a 100 mL volumetric flask. Afterwards, 0.7 g of each sample was submerged into a glass beaker consisting of 70 mL of 10 ppm MB and immediately exposed to sunlight for a total of 300 min. Every 30 min, about 40 mL of the irradiated solution was transferred into a glass vial with a dropper to investigate its degradation behavior using an ultraviolet-visible (UV-Vis) spectrophotometer, (Lambda 25, Perkin Elmer, Waltham, MA, USA) with 650 nm of wavelength. The degradation ratio of MB was determined using Equation (4); where *A_t_* is the degradation ratio, *I*_0_ is base absorbance, and *I_t_* is absorbency after time *t* [33,73,75].
(4)$$A_{t}=\frac{I_{0}-I_{t}}{I_{0}}\times 100$$

### 2.6. Characterizations

To identify chemical bonds and to locate the functional group of the cellulose films, the samples were characterized using a Fourier-transform infrared (FTIR) analysis. The analysis was carried out using an ATR-Perkin Elmer Spectrometer 2000 FT-IR, based on the attenuated total reflection (ATR) approach to ensure non-destructive analysis. The measurement was conducted under 650 cm^−1^–4000 cm^−1^ and 4 cm^−1^ resolution. The phase determination and crystallite size measurement were conducted using a Rigaku RINT 2000 X-ray diffraction (XRD). The reflection mode with monochromator-filtered Cu Kα radiation (λ = 0.15418 nm) at 30 kV and 10 mA was applied. Samples were prepared in two different ways depending on the type of tested sample. For the powder sample, it was loaded onto the XRD sample holder with a small circle cavity in the middle of the holder. For film samples, the cellulose film was cut into a size of 25 mm, thickness ≤ 8 mm, and stuck on the holder. The prepared samples were subsequently scanned with 2theta in the range between 15° and 80°. The Debye-Scherrer Equation (5) was generated to compute the crystallite size and the degree of crystallinity was obtained from Equation (6) as follows:(5)$$D=\frac{K\lambda }{\beta \;\cos\theta }$$
(6)$$\mathrm{Degree}\;\mathrm{of}\;\mathrm{Crystallinity}=\frac{\mathrm{Area}\;\mathrm{of}\;\mathrm{all}\;\mathrm{crystalline}\;\mathrm{peaks}}{\mathrm{Total}\;\mathrm{area}\;\mathrm{under}\;\mathrm{XRD}\;\mathrm{peaks}}\times 100$$
where, D = crystallite size (nm); K = Scherrer’s constant (K = 0.94); λ = the wavelength of X-ray (1.54178 Å); β = full width half maximum (FWHM); θ = angle of diffraction (rad) [76].

## 3. Results and Discussion

### 3.1. The Formation of Cellulose-Photocatalyst Film

In nature, the structure of cellulose includes both axial C–H bonds and equatorial hydroxyl groups. Hence, it has amphiphilic properties with the inter-layer region being hydrophobic and the intra-layer region being hydrophilic. The intra- and intermolecular hydrogen bonds should be broken to a large extent when the cellulose is dissolved in the solvent [77]. Thus, when cellulose is dissolved in the NaOH/urea solution, it is expected that the NaOH interacts with the hydrophilic hydroxyl groups of cellulose, breaking the hydrogen bond between the chains, and the urea reacts with the hydrophobic portions, resulting in a solution that dissolves cellulose [78].

In this study, it was also discovered that the semi-crystalline properties of the cellulose can impact the end-product of the prepared films. During the drying process, the cellulose films will experience severe shrinkage [72,79,80]. Nevertheless, when 2.5% *w*/*v* glycerol and 5% PVA bath was applied, the shrinkage of the film samples became less. This is because the existence of PVA as a plasticizer in the film can improve the toughness of the cellulose glycerol composite films [73]. The commercial Ag_2_S powder was micron- sized and might affect the distribution of particles when incorporated in the MCC film. Thus, the rarefaction process was conducted as shown in Figure 2.

The particle size transformations were involved during the process. The stirring action from the magnetic stirrer was deployed in a clockwise direction at a certain speed in the liquid while applying fluidic shear forces among the involved substances [81]. The existence of opposing forces between the applied forces and the friction forces within the system (Figure 2) caused the occurrence of shear stress in the Ag_2_S powders, which can reduce the Ag_2_S particle size. The longer the magnetic bar rotates, the more forces work, resulting in more energy provided to wear down the Ag_2_S powder. Thus, smaller crystal Ag_2_S were formed. According to the peak analysis and crystallite size calculation, it was confirmed that before stirring, the Ag_2_S particle has a crystallite size of 52 nm and this size was reduced to 10 nm after the rarefaction process. Thus, the purpose of the rarefaction in this study was succeeded.

The distribution of both the rarefacted and synthesized Ag_2_S powder in the cellulose film can be observed in Figure 3. According to Figure 3d–f, the particles from the synthesized Ag_2_S particles evenly spread in the whole area of the cellulose matrix. Meanwhile, the samples that were loaded with the commercial Ag_2_S had diverse particle distributions (Figure 3a–c). It was shown that part of the cellulose film was not affected by the Ag_2_S. The degree of crystallinity and the crystallite size of the synthesized Ag_2_S can be categorized as one of the factors that affected this particle distribution. Therefore, further discussion about the structure and chemical function of the samples will be explained in the following section.

### 3.2. Crystalline Phase Investigation of the Synthesized Ag_2_S powder

Figure 4 shows the XRD spectra of the synthesized Ag_2_S particles that were obtained from the reaction of 0.3 M AgNO_3_ and thiourea. The diffraction peaks of Ag_2_S that were observed at 2theta = 22.4°, 25.9°, 28.9°, 31.5°, 34.4°, 36.8°, 37.7°, 40.7°, 43.6°, and 46.2° showed similar patterns with the commercial Ag_2_S and confirmed the formation of Ag_2_S phase (ICDD No. 00-014-0072) with monoclinic structure.

### 3.3. Crystal Structure and Phase Identification of Cellulose Film/Synthesized Ag_2_S (CF/syntAg_2_S)

To ensure the formation of the synthesized silver sulfide in the cellulose film samples, it was necessary to delve into the characterization of their crystal structure and phase identification. Furthermore, the influence of the AgNO_3_ concentration on the crystal structure of the sample was also studied. The crystallite is crucial because the crystallinity of the catalyst in the film will have a direct impact on the photocatalytic properties.

The study of the crystal structure of the CF/syntAg_2_S sample confirmed that all as-prepared samples have diffraction peaks of a monoclinic Acanthine Ag_2_S structure with space group P_21_/C. The specific lattice parameter of these samples can be indexed as a = 9.5200 Å, b = 6.9300 Å, and c = 8.2900 Å. Moreover, the peak of (002) due to the reflection of the cellulose plane also existed in all of the samples that had undergone different AgNO_3_/thiourea loadings during the synthesis. The acquired peaks of the Ag_2_S and cellulose were matched well with the references ICDD No. 00-009-0422 and ICDD No. 00-050-2241, respectively. The average grain size and degree of crystallinity of the samples are shown in Table 4.

In terms of peak intensity, it was observed that the higher the concentration, the lower and broader the peak (see Figure 5), which is attributed to the low crystallite size of the sample [82]. Table 4 illustrates that the smallest yield of crystallite size was found from the highest concentration of the substituent (CF/syntAg_2_S). Furthermore, Table 4 also depicts that the substituent was responsible for the shifted peak that occurred in the sample with concentrated loadings. The shifted peak towards the lower 2theta indicated that the lattice parameter of the pristine cellulose had been incorporated by the synthesized Ag_2_S [83,84].

### 3.4. Functional Group Analysis of CF/comAg_2_S and CF/syntAg_2_S

According to Figure 6, the commercial (Figure 6a–d) and synthesized (Figure 6g–j) samples had comparable FTIR patterns along the observation scanning range. However, in terms of peak intensity, the sample with the synthesized Ag_2_S had a sharp and narrow peak compared to the commercial sample. The spectra illustrated the same pattern among the thin films that were either impregnated by the commercial or synthesized Ag_2_S. The broad bands that were found from wavenumber 3336.5 cm^−1^ to 3419.8 cm^−1^ were designated to the hydroxyl (O-H) group stretching vibration. The shifted peak among these bands was attributed to the interaction of the hydrogen bond of the pristine cellulose with glycerol and PVA. These displacements were also found by [71,85], in which the high content of glycerol and PVA in the cellulose matrix impacted the sharpening and shifting to the higher wavenumber. The spectral band located from 2916.5 cm^−1^ to 2912.59 cm^−1^, as well as a weak peak at 897.00 cm^−1^ were attributed to the characteristic of symmetrical stretching of C-H from the alkyl group [32] and glycosidic CH deformation [72], respectively.

Thereafter, the transmittance band at the area of 1753.70 cm^−1^ to 1752.13 cm^−1^ was associated with the stretching vibration of the C=O ester carbonyl group and the medium peak at the range of 1238 cm^−1^ was denoted as C-O-C asymmetric stretching [86]. The authentic peak of cellulose was found at the region around 1059 cm^−1^ and remarked as C-O-C stretching vibrations of aliphatic primary and secondary alcohols in cellulose [87,88]. The peak of C-H bending was observed at the regions of 1374 cm^−1^ to 1373 cm^−1^. The study by Cazón et al. announced that the chemical content of glycerol and PVA in the cellulose matrix may contribute to the displacement of the observed peak. Nevertheless, Figure 6e,f,k,l indicates that the loadings of metal silver sulfide into the cellulose can reduce the intensity of the peak with an oxygen-containing functional group. The more concentrated the Ag_2_S, the lower the intensity of the peak-contained oxygen functional group. According to Kumar et al., it was confirmed that the strong electrostatic linkages between Ag_2_S and the functional groups of cellulose were responsible for the lower intensity of the transmittance peak into the oxygen-containing functional group [89]. This means that the different intensity of the peak, particularly in the range of oxygen-containing group regions, was a characteristic that showed the success of Ag_2_S deposited into the cellulose matrix.

### 3.5. Photocatalytic Degradation Mechanism of Methylene Blue Using the Cellulose/Ag_2_S Films

The degradation of the methylene blue (MB) dye solution with sunlight exposure and without exposure was investigated. The results are shown in Table 5. For the non-exposure samples, the degradation rate of both synthesized and commercial Ag_2_S incorporated in the cellulose film was only 35% and 33% (due to the adsorption by cellulose film), respectively, even after 5 h of processing. Compared to the same samples that were exposed to sunlight, the degradation rate of MB reached 98–100% within 2 h. This result showed that solar energy seemed to have a great influence on the photocatalytic activity of the Ag_2_S-containing samples, since this semiconductor catalyst absorbed in the NIR region. Nevertheless, according to the direct observation of photocatalytic activity under sunlight exposure, there were two kinds of reactions that might have occurred: firstly, the reaction between the cellulose surface and the MB solution; secondly, the reaction among deposited Ag_2_S in the cellulose film and the MB solution. Figure 7 shows the reaction mechanism that might have occurred between the cellulose and MB solution during the process.

The cellulose interacts with MB through several bonds, such as the electrostatic bond between the ion N^+^ in MB and ion O^−^ in cellulose. The electrostatic attraction that was involved in the adsorption mechanism of MB on cellulose led to the enhancement of the MB molecules to quickly fill the adsorption sites on the surface of the cellulose film, resulting in a high rate of MB adsorption [90]. The Van der Waals force (C=C) and hydrogen bond also formed during this process [91]. These bonds made it possible for the cellulose to experience the photolysis reaction when exposed to the solar source.

On the other hand, under solar illumination, the existence of the catalyst Ag_2_S in the cellulose film established the photocatalytic activity due to the configuration of the reducing and oxidizing (electron–hole pairs) ions on the surface of the catalyst, which may degrade the concentration of the MB dyes solution until it reached 100% discoloration (see Table 5). During the process of solar irradiation, photons of solar light with an energy that was either equal to or higher than the band gap of Ag_2_S (~1.06 eV) [92] were degraded by Ag_2_S. Upon the process of photon absorption, an electron (*e^-^*) from the valence band (VB) was stimulated up to the conduction band (CB). In addition, it was linked to the development of a hole (*h^+^*) in the valence band, which led to the formation of electron–hole pairs that took part in the process of reduction and oxidation. In this process, the electrons in the surface reacted with the oxygen (O_2_) that was dissolved in the aqueous solution, which then resulted in the production of anionic superoxide radicals (•O_2_^−^) [93]. Simultaneously, the photogenerated holes also reacted with water to create hydroxyl radicals, which further oxidized the dye molecules. The produced anionic superoxide radicals (•O_2_^−^) and hydroxyl radicals oxidized the dye molecules that can decompose the MB dyes into CO_2_ or H_2_O and their intermediates [94]. Overall, this phenomenon can be expressed through the following chemical reactions:Ag_2_S + *hv* → *e*^−^ + *h^+^*(7)
*e*^−^ + O_2_ → •O_2_^−^(8)
*h^+^* + H_2_O → •OH + H^+^(9)
(10)$$\mathrm{Methylene}\;\mathrm{Blue}\;\to \;\overset{\mathrm{free}\;\mathrm{radicals}}{\overbrace{•O_{2}{}^{-}•\mathrm{OH}\;}}\;\to \;\mathrm{intermediates}\;+\;{\mathrm{CO}}_{2}\;+\;H_{2}O$$

However, without light energy, the processes involving reduction–oxidation reactions in the photocatalysis process will not be formed. As a result, the MB dyes wastewater will only interact with the cellulose film as described above.

The incorporation of Ag_2_S into cellulose films served primarily to increase the rate of dye degradation by creating a massive contact surface. The catalyst, Ag_2_S, was distributed well toward the cellulose films, leading to the films outperforming the powder counterparts in terms of activity because they effectively absorbed light and perhaps underwent internal scattering within the cellulose/Ag_2_S film, inducing higher charge carrier formation and hence better photocatalytic efficiency. Table 5 shows the photocatalytic efficiency for all samples based on the MB degradation rate (%). From the table, when Ag_2_S powder was used, the degradation rate of MB only reached about 75% even after 5 h of sunlight exposure. For pristine cellulose, the degradation of MB reached 100% after 4 h exposure. The high rate of MB adsorption by pristine cellulose might be due to the electrostatic attraction that was involved in the adsorption mechanism of the MB molecules to quickly fill the adsorption sites on the surface of the cellulose [90]. For CF/com Ag_2_S, 100% of the MB was degraded in 2 h of exposure time compared to the CF/syntAg_2_S sample with 98.6% degradation. The deterioration in the treatment that applied the catalyst was compared to the degradation in the photolysis experiment that was also conducted without the presence of a catalyst in the aqueous solution of methylene blue. By observing the direct photolysis of MB, it was clear that there were no appreciable color changes over the 5 h of exposure time. The degradation rate also reached its lowest value, ranging from 0% to 1.66%, as shown in Figure 8a,c.

Based on Figure 8b,d, the concentration of the dye solution was elevated to more than 90% in just 60 min for the pristine cellulose, CF/comAg_2_S and CF/syntAg_2_S samples, while for the Ag_2_S powder, less than 40% of the MB was degraded. This showed that the distribution of Ag_2_S powders in regenerated cellulose matrices can enhance the photocatalytic activity up to 100% efficiency due to higher surface area, and the affinity of the photocatalyst films that react with the dye molecules during the sunlight irradiation [95].

In addition, the surface of the cellulose film acted as a host during the process of dye adsorption, which in turn enabled the donor and acceptor molecules to interact with one another. This process could delay the recombination of charge carriers under direct sunlight. Charge recombination must be avoided during this process since it can bring down the efficiency of the photocatalyst [96,97,98]. In terms of the degradation ratio, it can be seen that there were differences in the photodegradation characterization of the two types of samples that were synthesized. The samples with commercial Ag_2_S doping will have an increasing ability with increasing Ag_2_S loading, where the higher content that can be accepted by cellulose film is 0.1 g. Meanwhile, the other types of samples (CF/syntAg_2_S) showed the opposite nature. This may occur due to overload on the sample (Figure 3d–f) with loading that might exceed the limit, resulting in poor interfacial charge carrier migration and an increase in the recombination rate of photoexcited electron–hole pairs due to an increase in the concentration of Ag sources [73,74]. Thus, it was found that the CF/comAg_2_S samples had the efficiency to degrade 100% of the MB content compared to the CF/syntAg_2_S (98.6%) in 2 h.

## 4. Conclusions

It has been successfully demonstrated that silver sulfide (Ag_2_S) particles, which were incorporated into regenerated cellulose, outperformed its properties in the powder state. This was because the distribution of Ag_2_S particles on the cellulose matrix could increase its photocatalytic activity as the contact surface between the catalyst and dyes became larger. Furthermore, the molecules contained in the cellulose were found to have the ability to interact with molecules in the dyes, especially the methylene blue (MB) dyes. The samples with the commercial Ag_2_S showed an increase in photocatalytic activity up to 100% degradation after 120 min of sunlight exposure with the maximum of 0.1 g Ag_2_S loading onto the cellulose film, while other types of samples (CF/syntAg_2_S) showed 98.6%. This situation might be due to the overload of synthesized Ag_2_S particles on the sample that exceeded the photocatalyst limit on the film. As a result, poor interfacial charge carrier migration and an increase in the recombination rate of photoexcited electron–hole pairs occurred. Thus, it can be concluded that the photocatalytic efficiency of the photocatalyst film originating from the commercial Ag_2_S particles was the highest, with 100% of the degradation rates in 2 h. For the non-exposure sunlight samples, the degradation rates were only 33–35%, showing the importance of the NIR semiconductor Ag_2_S catalyst used.

## Figures and Tables

**Figure 1 materials-16-00437-f001:**
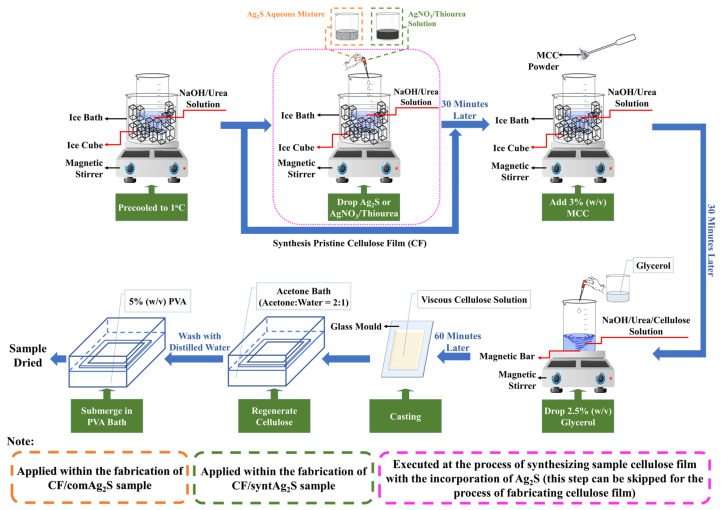
Schematic illustration of sequence process to synthesize the samples.

**Figure 2 materials-16-00437-f002:**
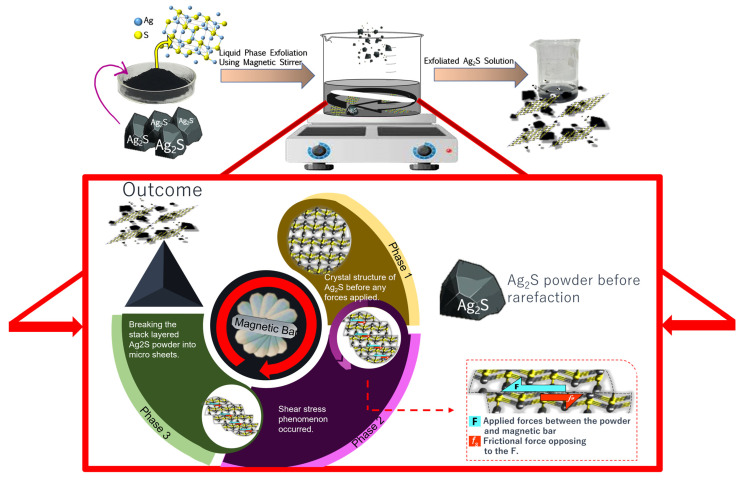
Illustrated scheme of the rarefaction process of commercial semiconductor Ag_2_S.

**Figure 3 materials-16-00437-f003:**
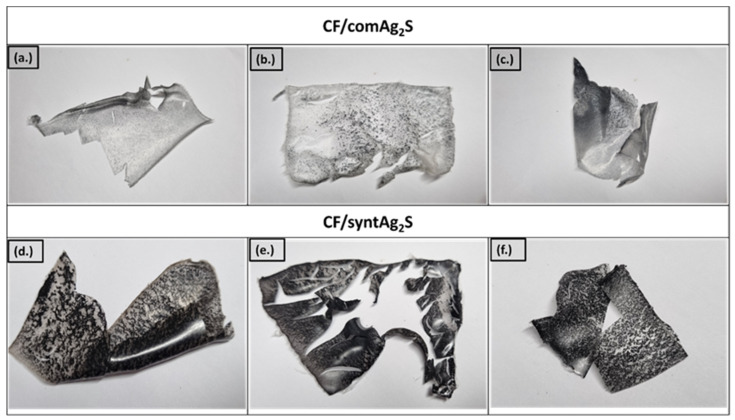
Prepared cellulose films: (**a**) CF/comAg_2_S1, (**b**) CF/comAg_2_S2, (**c**) CF/comAg_2_S3, (**d**) CF/syntAg_2_S1, (**e**) CF/syntAg_2_S2, and (**f**) CF/syntAg_2_S3.

**Figure 4 materials-16-00437-f004:**
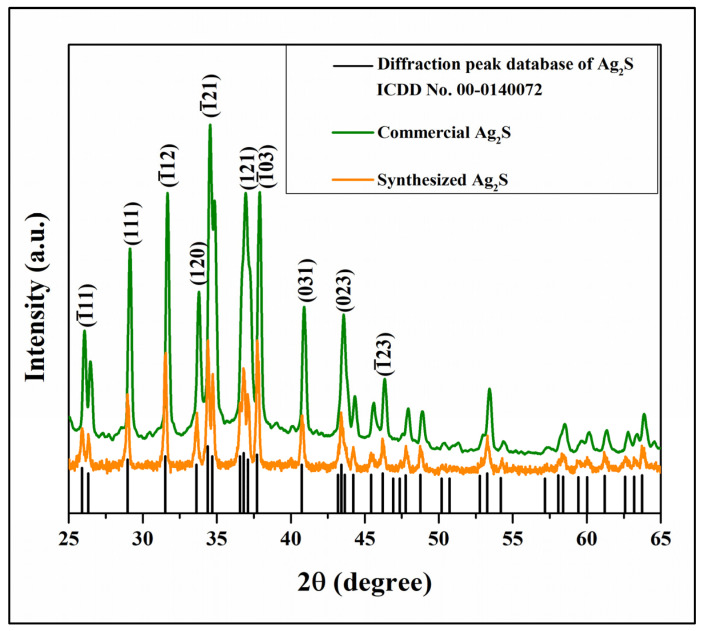
X-ray diffraction spectrum of synthesized Ag_2_S.

**Figure 5 materials-16-00437-f005:**
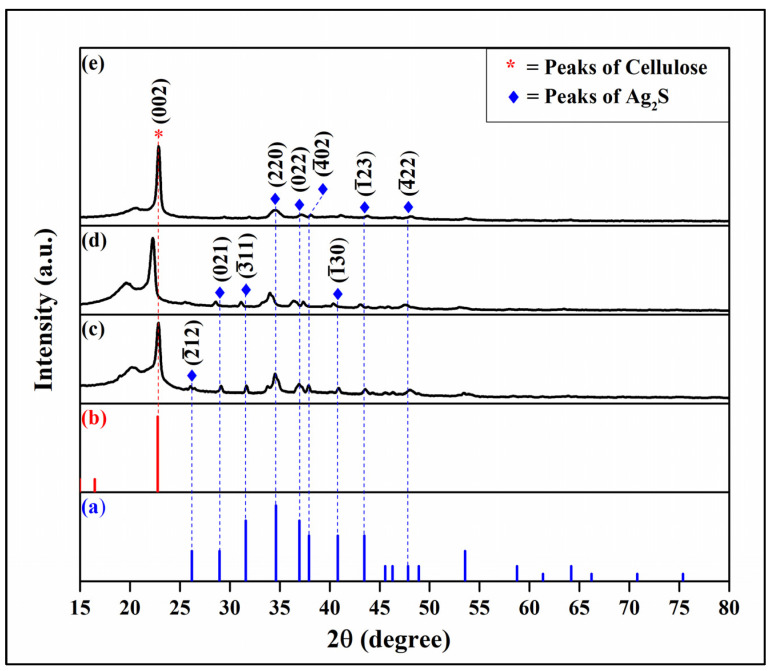
X-ray diffraction spectra of: (**a**) database of Ag_2_S (ICDD No. 00-009-0422), (**b**) database of cellulose (ICDD No. 00-050-2241), (**c**) CF/syntAg_2_S1, (**d**) CF/syntAg_2_S2, and (**e**) CF/syntAg_2_S3.

**Figure 6 materials-16-00437-f006:**
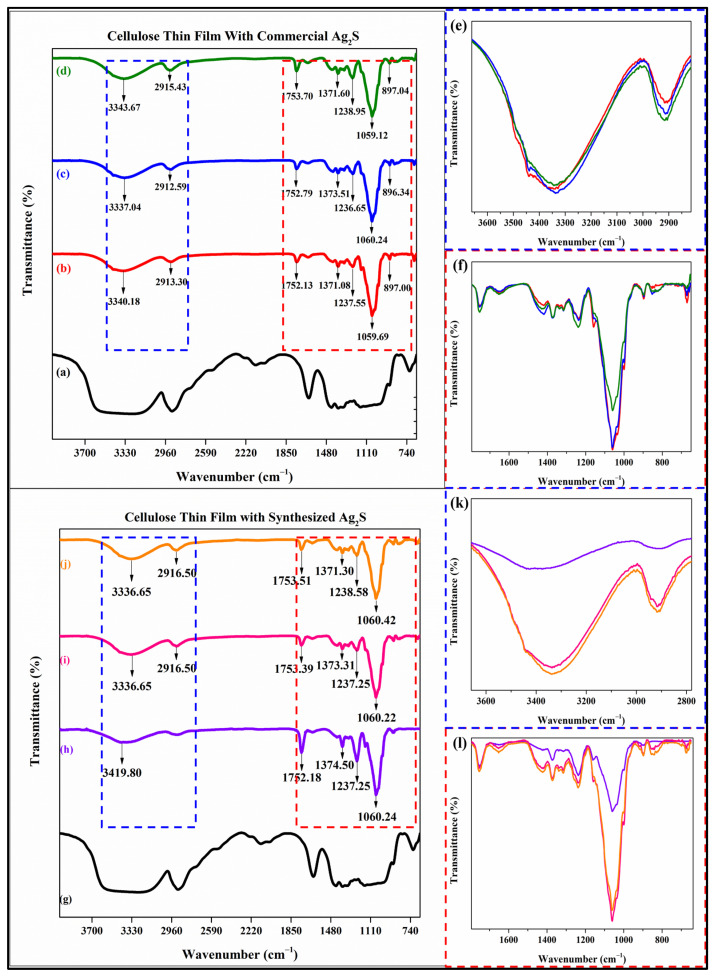
ATR—FTIR spectra of: (**a**,**g**) cellulose, (**b**) CF/comAg_2_S1, (**c**) CF/comAg_2_S2, (**d**) CF/comAg_2_S3, (**e**,**f**) inset CF/comAg_2_S graph, (**h**) CF/syntAg_2_S1, (**i**) CF/syntAg_2_S2, (**j**) CF/syntAg_2_S3, and (**k**,**l**) inset CF/syntAg_2_S graph.

**Figure 7 materials-16-00437-f007:**
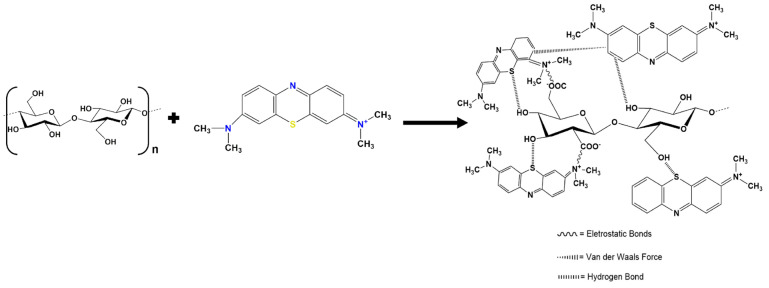
Predicted mechanism reaction between methylene blue and cellulose.

**Figure 8 materials-16-00437-f008:**
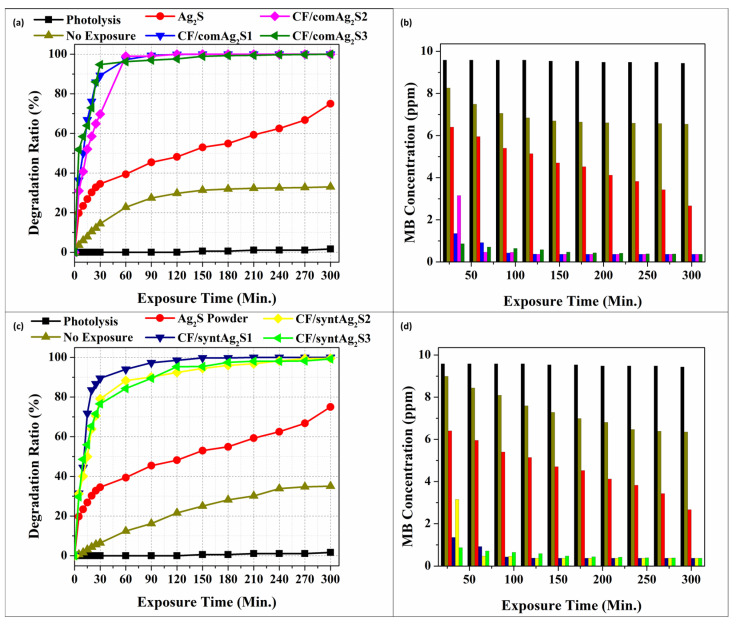
(**a**) Degradation ratio of MB with Ag_2_S commercial as a catalyst; (**b**) The concentration of MB after photocatalytic activity using CF/comAg_2_S samples; (**c**) Degradation ratio of MB with synthesized Ag_2_S as a catalyst; (**d**) The concentration of MB after photocatalytic activity employing CF/syntAg_2_S samples.

**Table 1 materials-16-00437-t001:** Comparison of the cost-effectiveness assessment of several photocatalysts.

Type of Photocatalyst	Types of Wastewater	Energy Sources	Catalyst Dosage (g/L)	Maximum Photodegradation Efficiency	Time to Obtain Maximum Photodegradation (min.)	Total Power Consumed (kWh)	Operational Cost (USD/kg) ^b^	Refs.
Rutile TiO_2_	Acetaldehyde	UV lamps (56 W)	5.01	80	100	not available	127.13	[64]
GAC ^a^–TiO_2_	Livestock wastewater	UV lamps (56 W)	6	100	6	0.524	0.68	[65]
V_2_O_5_–TiO_2_	Methylene blue	UV lamps	4.75	92	120	1	8.7205	[66]
TiO_2_/O_3_	Tert-butyl alcohol	UVA lamps (15W)	not available	75	10	37	not available	[67]
Cellulose/Ag_2_S	Methylene blue	Solar energy	7	100	120	0	0	Current study

^a^ Granular activated carbon. ^b^ Energy costs (USD) per kg of wastewater removal.

**Table 2 materials-16-00437-t002:** Sample code and composition of refracted commercial Ag_2_S.

Sample Code	Ag_2_S Mass (Gram)	Volume of Distilled Water (mL)
CF/comAg_2_S1	0.075	6.0
CF/comAg_2_S2	0.100	6.0
CF/comAg_2_S3	0.300	6.0

**Table 4 materials-16-00437-t004:** Phase identification and crystallinity of the CF/syntAg_2_S sample.

Sample Code	Index Miller Cellulose		Index Miller Ag_2_S						Crystallite Size (Å)	Degree of Crystallinity (%)
	(002)		$\left(\bar{3}11\right)$		(220)		(022)			
	2θ Stnd.	2θ Ob.	2θ Stnd.	2θ Ob.	2θ Stnd.	2θ Ob.	2θ Stnd.	2θ Ob.		
CF/syntAg_2_S1	22.78	22.76	31.56	-	34.50	34.44	36.92	37.18	221	47.93
CF/syntAg_2_S2	22.78	22.30	31.56	31.18	34.50	33.99	36.92	36.47	188	76.76
CF/syntAg_2_S3	22.78	22.78	31.56	31.48	34.50	34.49	36.92	36.99	174	37.75

**Table 5 materials-16-00437-t005:** Comparison of the photocatalytic efficiency of the prepared samples.

Exposure Time (Min)	Degradation Ratio (%)					
	Non-Exposure		Ag_2_S Powder	Pristine Cellulose	With Exposure	
	CF/comAg_2_S	CF/syntAg_2_S			CF/comAg_2_S2	CF/syntAg_2_S1
0	0	0	0	0	0	0
5	3.59	0.66	19.78	44.14	30.99	31.49
10	5.97	1.60	23.37	53.09	40.77	44.42
15	7.84	3.09	26.85	63.43	52.10	71.71
20	10.44	4.31	30.22	71.10	58.56	83.54
25	12.21	5.64	32.82	75.91	64.92	86.69
30	14.42	6.46	34.53	81.66	69.78	89.45
60	22.76	12.43	39.39	90.88	98.95	93.98
90	27.40	16.24	45.41	94.92	99.01	97.29
120	29.78	21.60	48.18	96.19	100	98.56
150	31.33	25.03	52.98	99.56	100	99.78
180	31.93	28.18	54.92	99.56	100	99.78
210	32.32	30.17	59.28	99.83	100	100
240	32.49	33.81	62.49	99.97	100	100
270	32.71	34.75	66.74	100	100	100
300	33.04	35.08	75.03	100	100	100

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
