# Peer review of "Near-Infrared (NIR) Silver Sulfide (Ag2S) Semiconductor Photocatalyst Film for Degradation of Methylene Blue Solution"

_materials, 2023, doi:10.3390/ma16010437_

Round 1

Reviewer 1 Report

1. The language of this paper needs further polishing. There are many confusing and inaccurate descriptions and comments across the whole paper. Some are listed here.

1) ‘Photocatalysis is an accelerated chemical reaction that is aided by light and semiconductor catalyst to eliminate dye pollutants from aqueous media.’ This is definitely not the definition of photocatalysis. Semiconductor is not necessary and dye degradation is not the only application.

2) ‘Commonly, photocatalysis exploits powder forms of metal oxide as semiconductor elements, but those are not without certain drawbacks in terms of cost, performance, and retrieval.’ ‘Those are not without certain drawbacks’ needs revision.

3) Some sentences use ‘cellulose film’ while some use ‘films’, which need to be aligned.

4) ‘In the midst of AOPs, heterogenous photocatalysis has been considered as a leading 75

method and desired breakthrough to cope with both aquatic organic contaminants.’  ‘Both aquatic organic contaminants’ need revision. If the authors wanted to claim both aquatic and organic contaminants. These two are not two categories and do have overlaps.

5) ‘The most common technique in photodegradation is by utilizing the powder form of photocatalyst, such as Ag2S powders’. Utilizing powder is not sort of technique….

2. The vision of this paper is not clearly explained. In the abstract and introduction, the authors stated that semiconductors have drawbacks like high cost, hard to recycle, poor performance and then Ag2S was picked but didn’t explain why Ag2S is outstanding except for the narrow band gap. Also, the electron mobility or charge separation efficiency of the Ag2S was not mentioned. Narrow band gap may not be the advantage as narrow band gap semiconductors could have low charge separation efficiency, which may hurt the photocatalysis performance a lot.

3. MB was chosen to represent dye molecules but in the intro and abstract, the authors didn’t explain why. Different dyes have different oxidation or reduction pathways.

4. The authors mentioned cellulose films were used because they are eco-friendly but didn’t explain whether Ag2S can cause pollution. Semiconductor nanoparticles are generally harmful to the aqua life.

5. Fig 4 may need scale bar to show the various amount of material is not the root cause of different performance.

Author Response

The respond to Reviewer 1 as attached.

Reviewer 2 Report

This manuscript cannot be published in journal Materials at the present form and major revisions are necessary.

1/The authors should include the novelty of the work in the term of cost, reliability, and performance.

2/Abstract should contain more quantitative information, i.e., optimum conditions…

3/Why have you selected Methylene Blue and Silver Sulfide (Ag2S) for this study, why you don't choose other semiconductors? Mention the reason.

4/The authors are advised to explain how the developed materials will be cost effective and present a comparative table with others.

5/It is preferred to characterize the prepared composite material using XPS to investigate the interaction between silver sulfide (Ag2S) and cellulose.

6/ Why the authors do not study the optical properties of the synthesized composite?

7 / The authors should calculate the conduction band and valence band of Ag2S

8/ the authors should study the scavengers test. and then investigated the mechanism photocatalytic degradation.

9/Manuscripts should refer to and cite as much as possible from the last five years. Some high-quality literatures about photocatalyst and sustainability of water in recent years can be referenced and cited, such as:

https://doi.org/10.1007/s13369-022-06899-y;

https://doi.org/10.1016/j.seppur.2021.119399

Author Response

The respond to Reviewer 2 as attached.

Round 2

Reviewer 1 Report

The answers are satisfying.

Author Response

The English checking has been conducted.

Reviewer 2 Report

The revised version can be accepted for publication in this journal. 

Author Response

No further comments after Revision 1 submitted.